# British Columbia's Index of Multiple Deprivation for Community Health Service Areas

Sharon Relova [1], Yayuk Joffres [1], Drona Rasali [1,2,*], Li Rita Zhang [1], Geoffrey McKee [1,2] and Naveed Janjua [1,2]

[1]   British Columbia Centre for Disease Control (BCCDC), Provincial Health Services Authority, Vancouver, BC V5Z 4R4, Canada; sharon.relova@bccdc.ca (S.R.); yayuk.joffres@bccdc.ca (Y.J.); rita.zhang@bccdc.ca (L.R.Z.); geoffrey.mckee@bccdc.ca (G.M.); naveed.janjua@bccdc.ca (N.J.)

[2]   School of Population and Public Health, University of British Columbia, Vancouver, BC V6T 1Z3, Canada

*   Correspondence: drona.rasali@bccdc.ca

**Abstract:** Area-based socio-economic indicators, such as the Canadian Index of Multiple Deprivation (CIMD), have been used in equity analyses to inform strategies to improve needs-based, timely, and effective patient care and public health services to communities. The CIMD comprises four dimensions of deprivation: residential instability, economic dependency, ethno-cultural composition, and situational vulnerability. Using the CIMD methodology, the British Columbia Index of Multiple Deprivation (BCIMD) was developed to create indexes at the Community Health Services Area (CHSA) level in British Columbia (BC). BCIMD indexes are reported by quintiles, where quintile 1 represents the least deprived (or ethno-culturally diverse), and quintile 5 is the most deprived (or diverse). Distinctive characteristics of a community can be captured using the BCIMD, where a given CHSA may have a high level of deprivation in one dimension and a low level of deprivation in another. The utility of this data as a surveillance tool to monitor population demography has been used to inform decision making in healthcare by stakeholders in the regional health authorities and governmental agencies. The data have also been linked to health care data, such as COVID-19 case incidence and vaccination coverage, to understand the epidemiology of disease burden through an equity lens.

**Dataset:** http://www.bccdc.ca/Our-Services-Site/Documents/BCIMD%20CHSA%202016%20PCA%20Scores.xlsx (accessed on 15 January 2022).

**Dataset License:** CC0.

**Keywords:** social determinants; social vulnerability; social inequalities; deprivation; health disparities; geography; area-based socio-economic indicators (ABSIs)

## 1. Introduction

People's health and well-being are influenced by where they live and work, their demographic characteristics, socio-economic status, and many other social and material factors [1]. These factors influence the distribution of health outcomes that manifest among population groups across geographic areas; however, such factors are not uniform across British Columbia (BC) [2,3]. There are certain subpopulations across demographic, geographic, and socio-economic groups that are disadvantaged and do not experience the same level of healthcare access and health status as the rest of the population [3]. Health disparities, defined as the preventable differences in the burden of disease, injury, violence, or opportunities to achieve optimal health that are experienced by socio-economic groups of the population [4], exist in many jurisdictions locally and internationally. Further, health equality refers to the access to, or distribution of, resources evenly among individuals, whereas equity is the fair access to, or distribution of, resources according to an individual's

needs [5,6]. Identifying and quantifying these measures are necessary to provide a more equitable approach to health care.

BC, much like other jurisdictions across Canada, has universal coverage in public health care systems with health service utilization encompassing clinical patient care (primary care, hospitalization, prescription drugs) and public health services covering all residents in the province. One major advantage of universal coverage is that all healthcare utilization transactions are recorded and are accumulated as administrative data that can be readily used for system performance measurement as well as for population health surveillance and assessment. A disadvantage to using administrative data is that they are collected for transactional purposes and not for research or statistical purposes. Secondary analysis using administrative data may not contain sufficient detail or population scope for research or surveillance questions, especially because administrative data lack socio-economic or behavioral information for the individuals.

To provide equitable, needs-based, timely, and effective patient care and public health services to the population across communities, appropriate data are needed to capture geographic, demographic, and socio-economic factors influencing health. While administrative healthcare utilization data are commonly used, these sources generally do not contain detailed demographic or socio-economic information that could be used to assess patient care and public health outcomes through an equity lens grounded in the determinants of health. While metrics to measure and track health inequities are needed to promote equitable and positive health outcomes, there is a need to have summary measures that capture several determinants of health and reflect inequities that exist in the complex systems within which we live.

Canadian data sources that are currently available for deriving such composite measures primarily rely on area-based data collected from the census, which can be aggregated by census Dissemination Area (DA). These composite measures are developed as deprivation indexes, for which theoretical underpinning was provided by Townsend in the mid-1980s [7]. Deprivation, as measured by the index, can occur in various areas of livelihoods, including food, housing, education, work, or social ties that are distinguished between economic dimensions that measure material deprivation and social interactions that measure social deprivation [8]. The concepts of these two forms of deprivation were the bases of the Canadian version of the material and social deprivation indexes in Quebec, which was developed to track social and health inequalities over time and space [9]. Likewise, in developing a deprivation index in Ontario, the research process examined the index as a measure of poverty, capturing its various dimensions, such as social isolation, which is not reflected by income alone, while the index complements income measures rather than replacing them [10]. These indexes are the composite measures of a number of related socio-economic variables that have been used to capture overall health equity concepts [11].

Currently, in Canada, a multitude of area-based socio-economic indexes (ABSI) are available to measure social and material deprivation and their association with disparate health outcomes. These include the Vancouver Area Neighbourhood Index (VANDIX) in BC, comprising seven census variables [12]; the Socio-Economic Factor Index (SEFI-2) in Manitoba, comprising four census variables [13]; the Canadian Marginalization Index (CAN-Marg) in Ontario, comprising four dimensions of marginalization derived from 18 census variables [14]; and the Pampalon Index in Quebec, comprising two dimensions derived from six census variables [15]. While these ABSIs are strongly associated with various health outcomes, no one index is considered superior [16]. The most commonly cited index, the Pampalon Index [16], uses a principal component analysis (PCA) to attribute weights to six different variables reduced to two components of deprivation. PCA is the method most often used in the derivation of Canadian area-based deprivation indexes. The first component of the Pampalon Index is referred to as material deprivation, given the education level, employment, and average income variables, are most heavily weighted. The second component is referred to as the social deprivation, including persons living

alone, persons separated, widowed or divorced, and single-parent families variables. While indexes with more components would account for a larger proportion of total variance in health outcomes, the Pampalon Index uses only two components for the purposes of simplicity and ease of interpretation. In several studies, the social component of the Pampalon index had a disconcertingly weak association with health outcomes [16]. Other indexes, such as CAN-Marg, instead choose the number of components after examining the variation explained by each component in contrast to the Pampalon Index, which retained a pre-determined number of components. The former produces a more robust and adaptable index that accounts for more variation in health outcomes.

More recently, following the foundational research leading to the development of CAN-Marg [14], the Canadian Centre for Justice Statistics at Statistics Canada developed area-based multiple indexes of deprivation and marginalization using data from the 2016 Census of Population. The Canadian Index of Multiple Deprivation (CIMD) comprises four dimensions: residential instability, economic dependency, ethno-cultural composition, and situational vulnerability. Indexes are produced at the national, regional (Atlantic and Prairies), and provincial (Quebec, Ontario, and British Columbia) levels [17].

Indexes of marginalization, such as the CIMD, are useful sources of data that provide insight to enrich the health-related data through an equity lens. While the CIMD is available for the entire country, each region/province/territory has its own unique characteristics, such as geographic, demographic, and socio-economic diversity, and challenges, such as access to health care services and socio-economic inequities across subpopulations [3]. A BC version of the CIMD scores, which are estimated from the Census data aggregated at the DA level, provides additional context to issues most relevant to British Columbians.

In 2011, Provincial Health Services Authority (PHSA) released a report emphasizing the need for BC's health system to take further action towards reducing health inequities through the design, organization, and management of their specialized province-wide programs and services [18]. The report led to the prioritization of indicators to measure health inequities and monitor their progress across geographic, demographic, and socio-economic dimensions [19,20]. The British Columbia Centre for Disease Control (BCCDC), a part of PHSA, subsequently developed material and social deprivation indexes using the proprietary source data called 2011 CensusPlus and following the Pampalon Index methodology developed by Institut National de Santé Publique du Quebéc [21]. These indexes were applied to select priority health inequity indicators. When the 2016 Canada Long Form Census data became available, the BCCDC adopted Statistics Canada's CIMD to create the BCIMD. While Census data are cross-sectional, the development of BCIMD enabled a broader scope of health equity analysis, using four dimensions of deprivation indexes as the stratifying variables for examination of equity among the indicators of interest.

The BC health care system, available to all BC residents, is comprised of two provincial and five regional health authorities within the BC Ministry of Health. The system has defined geographic areas for health services delivery to understand and address the needs of the population within local areas, which fall within the responsibility of the regional health authorities. At the request of the health sector stakeholders needing local data for health assessment, the BC Ministry of Health created new geographic jurisdictions called Community Health Services Areas (CHSAs) that were more granular than pre-existing local health areas (LHAs). Released in 2019, CHSAs aligned coterminous census boundaries and health boundaries, allowing for improved public health planning and programming at the local level. CHSAs are mutually exclusive and are an exhaustive classification of the total land area in BC [22]. Currently, BC's provincial, regional, and local health systems are adopting CHSAs (as opposed to the DA level, currently available through Statistics Canada) as the base jurisdictions with greater granularity in their data for health planning. Socio-economic data for health equity metrics are therefore required at the CHSA level for

public health planning and programming and for monitoring variations in health system performance measures across communities in BC.

The objective of this paper is to describe the methodological adaptation of the CIMD to create the BC Index of Multiple Deprivation (BCIMD) at the CHSA level and its prospective uses in the health care system.

## 2. Methods

### 2.1. Data Sources

Demographic data from the 2016 Census of Canada microdata were used as the primary data source. The BCIMD for CHSAs was developed following the Statistics Canada methodology [17]. The 2017 and 2018 boundary configurations for 218 CHSAs were used in the creation of the indexes.

### 2.2. Data Description

The data product developed from this study is presented in the data workbook containing the following:

2.2.1. Data Dictionary
- Variables
- Descriptions

2.2.2. BCIMD Notes and Dimensions
- Data Notes
- Indicator dimensions

2.2.3. BCIMD CHSA 2016 Data (Column Names)
- CHSA2018_NUM
- CHSA18_NAM
- Ethno_Cultural_Composition_Quint
- Ethno_Cultural_Composition_Score
- Economic_Dependency_Quint
- Economic_dependency_Scores
- Residential_Instability_Quint

### 2.3. Data Analysis

The BCIMD was developed on the foundation of Statistics Canada's CIMD. Using 17 Census Canada 2016 variables within the four dimensions of multiple deprivations created for the BCIMD, principal component analysis (PCA), with oblique (promax) rotation, was conducted, and a factor score was generated for each of the four dimensions for all CHSAs in BC. Table 1 shows the definition and corresponding indicators for the four dimensions of deprivation. Some indicators were reverse-coded; for example, the "proportion of dwellings that are owned" was coded to the "proportion of dwellings that are rented". The dimensions are ordered from left to right according to the highest percentage of principal component variance explained in the data: ethno-cultural composition (left) had the highest percentage of variance explained in the data while residential instability had the lowest percentage of variance (right).

Higher factor scores indicate greater deprivation, and lower scores indicate lower deprivation. Economic dependency had the largest range in the scores, while ethno-cultural composition had the smallest range. Additionally, the quintile values within ethno-cultural composition appeared to be clustered, which may affect discrimination between quintiles (data not shown).

For ease of interpretation, each dimension was categorized into quintile rankings: quintile 1 for CHSAs that were the least deprived group in BC (or least diverse in the case of ethno-cultural composition), and quintile 5 for the most deprived (or most diverse in the case of ethno-cultural composition). Variability in quintiles for a given CHSA may exist: a CHSA may have some dimensions with quintile 1 (least deprived/diverse) and other dimensions with quintile 5 (most deprived/diverse), depending on the distinctive characteristics of that community.

Data are reported for 213 out of 218 CHSAs; the data for five CHSAs with small cell sizes were not provided by Statistics Canada due to privacy and confidentiality. All analyses were conducted and verified in SAS version 9.4 (SAS Institute, Carey, North Carolina, United States of America).

**Table 1.** Four dimensions of multiple deprivations in British Columbia, definition of deprivation, and indicators included.

| Dimension of Deprivation | Ethno-Cultural Composition | Situational Vulnerability | Economic Dependency | Residential Instability |
|---|---|---|---|---|
| **Concept captured at a British Columbia-level** | Diverse community composition of immigrant populations | Socio-demographic conditions in housing and education, and other relevant demographic characteristics | Participation in the labor force, or a dependence on other income sources besides employment income | Transient nature of neighborhood inhabitants, considering housing and familial factors |
| **Indicators included** | Proportion of population who self-identify as visible minority, the proportion of population that is foreign-born, the proportion of population with no knowledge of either official language (linguistic isolation), and the proportion of population who are recent immigrants (arrived in five years prior to Census) | Proportion of population that identifies as Aboriginal, the proportion of population aged 25–64 without a high school diploma, the proportion of dwellings needing major repairs, the proportion of population that is low-income, and the proportion of single-parent families. | Proportion of population participating in labor force (aged 15 and older)[1], the proportion of population aged 65 and older, the ratio of employment to population[1], and the dependency ratio (population aged 0–14 and aged 65 and older divided by population aged 15–64) | Proportion of dwellings that are apartment buildings, the proportion of persons living alone, the proportion of dwellings that are owned[1], and the proportion of the population who moved within the past five years |

[1] Indicators were reverse-coded.

## 3. Results

### 3.1. Data Products

The principal component factor analysis results consisting of the deprivation score and the quintile values derived from the score for each of the dimensions by CHSAs across BC are presented as outcomes of this study, except for five CHSAs, which had small population size requiring suppression of these data.

Quintiles and scores for each dimension for each CHSA are available for use by the public. Please see "Data Availability Statement" for more information.

Figure 1 illustrates the geospatial distribution for each quintile of the BCIMD dimension. In terms of ethno-cultural composition, the most diverse CHSAs are concentrated in the Lower Mainland within geographies where new immigrants often settle. In contrast, other CHSAs are considered relatively less diverse in this dimension, particularly in Northern BC. A similar pattern is observed for residential instability, where the most deprived CHSAs cluster around the high-mobility city centers, including Vancouver and Burnaby in the Lower Mainland, Victoria in Vancouver Island, and Kelowna in the Interior. CHSAs in Interior BC are among the most deprived in terms of economic dependency, as are several CHSAs on Vancouver Island. In contrast, CHSAs in Vancouver city core and the suburbs of Metro Vancouver are among the least deprived in this dimension. CHSAs in Vancouver city core and the suburbs of Metro Vancouver are also among the least deprived in situational vulnerability, as are those in and around Kelowna. The most deprived areas for this dimension are observed in northern Vancouver Island, parts of Interior BC, along the central coast, and extending to large parts of northern BC.

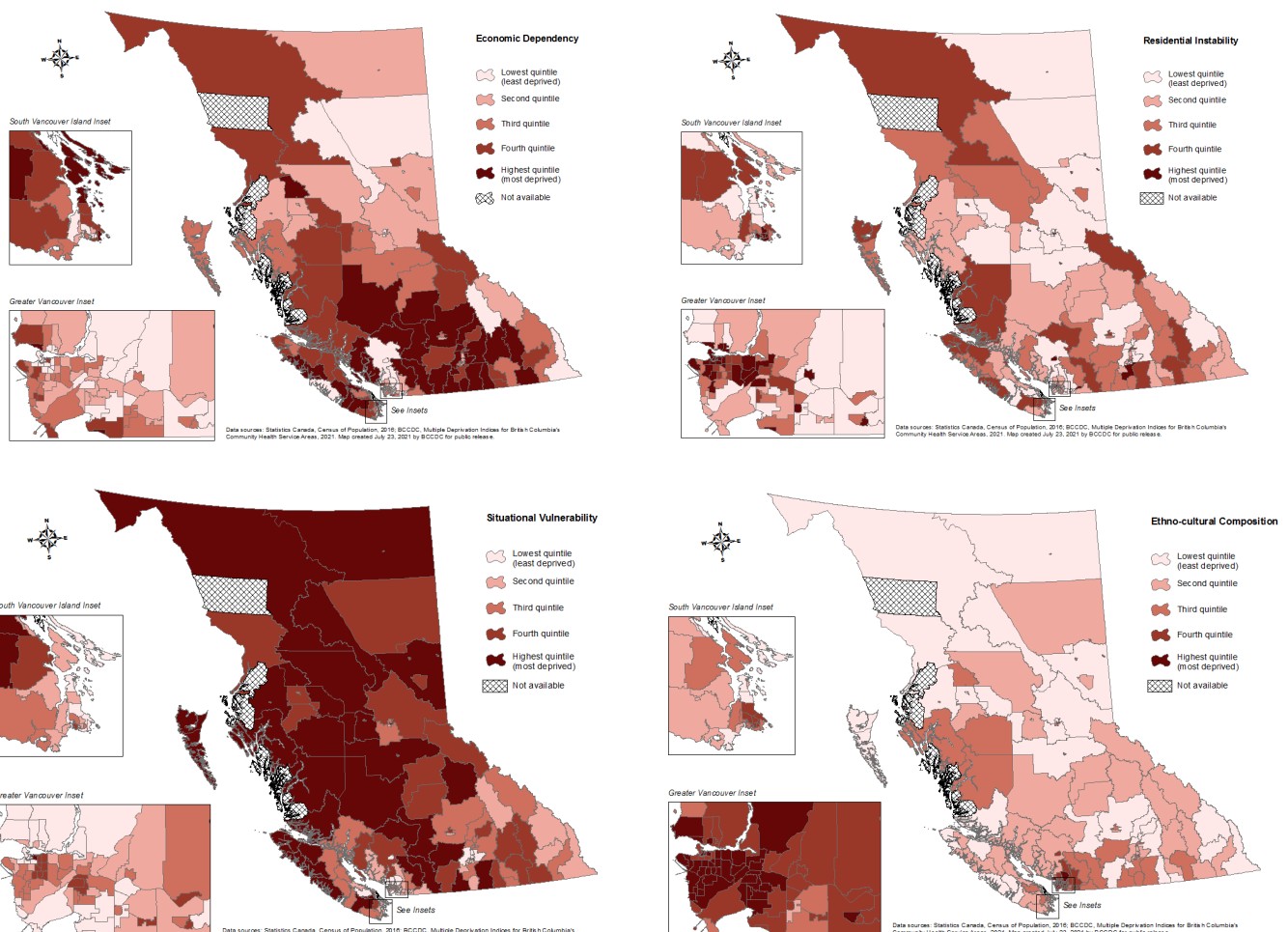

**Figure 1.** British Columbia's Indexes of Multiple Deprivations for community health service areas.

*3.2. Data Use*

The BCIMD has been utilized for a variety of purposes, including being a part of community health profiles for CHSAs throughout the province [23]. Deprivation indexes have also been provided to various stakeholders of the BC Public Health system and community partners. Stakeholders from regional health authorities, PHSA, and non-government organization partners link the BCIMD to health care data, such as COVID-19

case incidence, to better understand the epidemiology of disease burden from an equity perspective.

More recently, the BCCDC used the BCIMD to examine disparity in COVID-19 vaccination coverage in the province [24]. Biplot analyses of COVID vaccine coverage and BCIMD quintiles were conducted to assess disparity at a CHSA level and stratified for different adult age groups. For both 18–49 and 50+ year age groups, there was no clear pattern of one dose vaccine coverage observed by residential instability. There was a trend towards lower vaccine coverage in CHSAs with higher situational vulnerability and lower ethno-cultural diversity. CHSAs with a greater percentage of ethno-cultural diversity appeared to have greater vaccine coverage. There was an association between higher vaccine coverage in CHSAs with lower economic dependency among 18–49 years olds, but no association for 50+ year olds. This information is being used by BC provincial health leaders to inform their vaccination campaign strategies.

The BCIMD was also showcased in a provincial web mapping tool to enable data visualization and exploration of the distribution of COVID-19 cases and vaccination rates across BC. The tool is intended for authorized public health users, such as regional health authorities, BC Ministry of Health, and First Nations Health Authority, to view the distribution of cases, outbreaks, laboratory testing indicators, vaccine coverage, and additional human-social data layers that provide context and situational awareness. The BCIMD provided important contextual information to help interpret the data for decision-making purposes. The scores generated by the BCIMD are based on Census data and can change when the new cycle of Census is implemented. In Canada, the Census occurs every five years, and therefore, BCIMD scores are only valid within each Census cycle. Any variation in deprivation, if it exists, will be reflected between cycles.

## 4. Discussion

Research reported around the world confirms that area-based deprivation indexes are associated with equity in health status and health outcomes measured in terms of morbidity and mortality. Area-based deprivation was found to be associated with higher cancer mortality in Hong Kong, China [25] and with higher levels of anxiety among patients with advanced cancer in the United States [26], while a South Korean study suggested community deprivation levels to influence individual health behaviors [27]. A Northern Ireland study reported a significant correlation of the prescribing of multiple drug classes with socio-economic deprivation levels [28]. Another US study identified an association between spatial social polarization measured by deprivation index and risk of infant death, suggesting that efforts to support equitable community investments may reduce incidents of deaths in deprived areas [29]. A Canadian study reported an association of higher area-level material and social deprivation with higher rates of influenza-like illness-related Emergency Department visits [30]. Hence, the use of area-based deprivation indexes, such as BCIMD in the case of our study, can provide a feasible approach to identify disadvantaged communities or population groups with lower health status due to the nature of their inherent association.

### 4.1. Data Strengths

One of the greatest strengths of the BCIMD is that it accounts for a large proportion of variation in data at the CHSA level, given the high number of variables used to summarize each dimension. The results are consistent with Statistics Canada results [14] and make for a robust index that can capture a higher level of complexity compared to indexes with fewer, pre-determined components, such as the Pampalon Index.

In addition to serving as a standalone measure, another key strength of the BCIMD is that it enables linkages of area-based socio-economic diversity, deprivation, or marginalization information with ecologically corresponding health care utilization or health outcome data. This allows users to analyze and draw inferences on the association or impact of complex socio-economic factors on health and service-related outcomes [24]. By better

understanding these relationships, efforts to address health inequities may be better targeted within different geographic areas, acknowledging the unique characteristics of each population [23].

The BCIMD is an adaptable and robust tool. Boundaries for CHSAs are dynamic and can change depending on the needs of British Columbians. The BCIMD was initially developed using 2017 CHSA boundaries and revised accordingly when the 2018 CHSA boundaries were released. Comparison of the BCIMD scores between 2017 and 2018 CHSA boundaries showed similar results confirming the robustness of the index. Nevertheless, caution must be exercised when comparing BCIMD quintiles across different CHSA versions.

*4.2. Data Limitations*

The BCIMD is calculated at a CHSA level, and inferences may not hold true at the individual level. In particular, individual-level inferences must be based on individual-level data and should not be based on aggregate data, such as the BCIMD. This ecology fallacy is a limitation of all area-based socio-economic indicators and should be considered when interpreting results.

Misclassification of quintile assignment at the CHSA level can occur, and such misclassification is more likely to occur when a score is near the threshold between two quintiles. Because quintiles are derived based on a data-driven process of dividing the data into five equal proportions, the quintile assigned to one CHSA may not best reflect characteristics of that CHSA when compared to other CHSAs in that quintile, and instead, the other quintile ($+/-1$ quintile) may be more reflective of the CHSA. For example, a factor score of one CHSA may be on the border of quintiles 3 and 4. The CHSA was assigned a quintile 3 to ensure equal proportions of CHSAs within each quintile; however, the characteristics of the CHSA were more consistent with the CHSAs in quintile 4. The ability of BCIMD to discriminate between quintiles has not been described, and quintiles assigned to CHSAs can be misclassified. Discriminatory power is more important if there are many data points close to quintile cut-off values. Based on our analysis, however, the BCIMD quintile allocation is consistent with health status metrics for our population, as shown in our work on Community Health Profiles [23], so misclassification, if it exists, is not an issue.

Finally, like with all ABSIs, larger geographic areas are treated as homogenous and can lose the complexity of more granular geographic areas, which can mask some underlying inequities. Analysis of BCIMD scores by the category of urbanization (metropolitan/large urban/medium urban/small urban/rural hub/rural/remote) shows there are statistically significant geographic differences ($p < 0.05$) for each dimension: ethno-cultural diversity increases with increasing urbanization; economic dependency increases with decreasing urbanization; economic dependency and situational vulnerability decrease with decreasing urbanization; and residential instability increases with increasing urbanization, with the exception of remoteness—the latter most likely due to the temporal nature of resource development areas in BC (Table A1 and Figure A1).

In addition, because ABSIs use data based on residential addresses, they can also fail to capture social determinants of health associated with people's work or leisure environments. As such, caution should be used in over-interpreting results due to possible confounding effects of spatial autocorrelation.

**5. Conclusions**

The BCIMD is a useful area-based socio-economic indicator to quantify differences in socio-economic deprivation among local geographic areas in BC. Deprivation is measured by four dimensions: residential instability, economic dependency, ethno-cultural composition, and situational vulnerability. Within each dimension, quintiles are assigned to quantify the range of deprivation.

This paper outlines the development of the BCIMD, including data source, data description, and analysis. Results illustrate that deprivation is a multi-faceted concept and that CHSAs can have lower deprivation on one dimension and higher deprivation on another dimension. The quantification of the mix of these characteristics provides additional insight into the socio-economic situation of each CHSA.

In conclusion, as an area-based deprivation index that provides a feasible approach to identify the disadvantaged communities or the population groups with lower health status due to their inherent association, the BCIMD can be used to examine health inequities with the aim of providing evidence to support policy planning and evaluation, research and analysis, and resource allocation.

**Author Contributions:** Conceptualization: D.R., S.R. and Y.J.; methodology: Y.J., S.R. and D.R.; data analyses and curation: S.R., Y.J. and D.R.; geospatial maps L.R.Z.; writing—original draft preparation: S.R. and D.R.; writing—review and editing: all authors; supervision: D.R., N.J. and G.M.; project administration: D.R. and S.R. All authors have read and agreed to the published version of the manuscript.

**Funding:** This research received no external funding.

**Institutional Review Board Statement:** Not applicable.

**Informed Consent Statement:** Not applicable.

**Data Availability Statement:** The BCIMD data are available upon written request to pph@phsa.ca and are also publicly available for downloading directly by visiting the BCCDC website at http://www.bccdc.ca/our-services/programs/population-public-health-surveillance, (accessed on 15 January 2022).

**Acknowledgments:** The Canada Census 2016 data used for analysis in this study was obtained from Statistics Canada.

**Conflicts of Interest:** The authors declare no conflict of interest.

## Appendix A

**Table A1.** BCIMD and urbanization analysis of variance results.

| Source | Degrees of Freedom | Sum of Squares | Mean Square | F Value | Pr > F |
|---|---|---|---|---|---|
| Ethno-cultural composition | | | | | |
| Model | 6 | 273.4 | 45.6 | 61.5 | <0.0001 |
| Error | 206 | 152.6 | 0.7 | | |
| Corrected Total | 212 | 426.0 | | | |
| Residential instability | | | | | |
| Model | 6 | 75.4 | 12.6 | 7.4 | <0.0001 |
| Error | 206 | 350.6 | 1.7 | | |
| Corrected Total | 212 | 426.0 | | | |
| Economic dependency | | | | | |
| Model | 6 | 71.6 | 11.9 | 6.9 | <0.0001 |
| Error | 206 | 354.4 | 1.7 | | |
| Corrected Total | 212 | 426.0 | | | |
| Situational vulnerability | | | | | |
| Model | 6 | 109.9 | 18.3 | 11.9 | <0.0001 |
| Error | 206 | 316.1 | 1.5 | | |
| Corrected Total | 212 | 426.0 | | | |

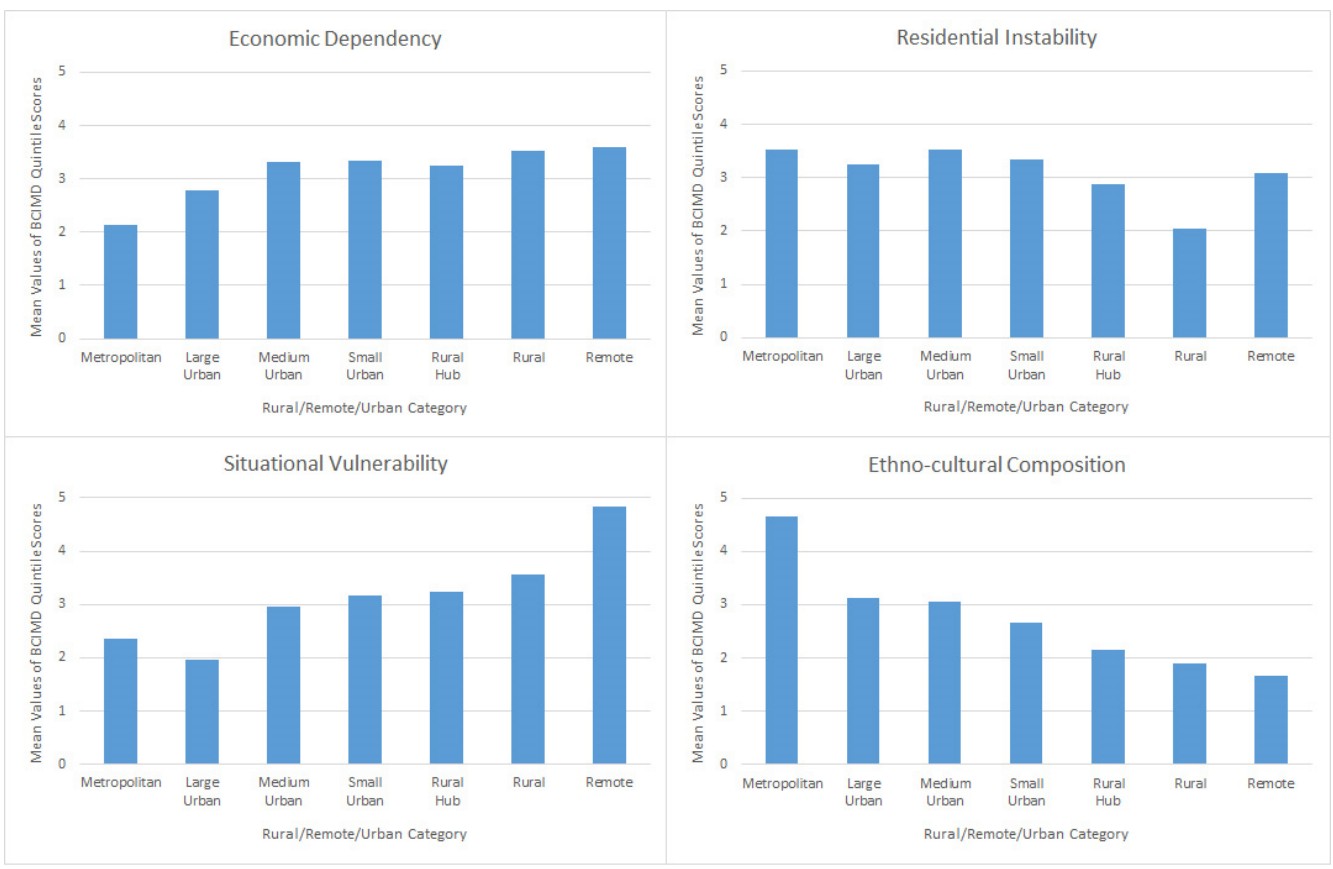

**Figure A1.** Mean values of British Columbia's Index of Multiple Deprivation for Community Health Service Areas (BCIMD) quintile scores by urbanization category.

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
