# Peer review of "British Columbia’s Index of Multiple Deprivation for Community Health Service Areas"

_data, 2018_

Round 1
Reviewer 1 Report
Overall, this is an excellent paper and I have very few comments to offer. I am not a Canadian scientist so any minor discrepancies in measures used should be debated by those with knowledge of those measures. The authors addressed my primary concerns of potential misclassification in the Data Limitations section. Ideally, there would be some sensitivity test here. This is beyond my personal methodological knowledge to specify the type of sensitivity test but some assessment of misclassification could induce bias would be welcome. Ideally, I would like to see stronger caution against ecological fallacy for less research savvy readers. Some recognition that cultural identities (i.e., self-identification) as measured can change over time within the same individual would also be welcome. This would have the potential to alter the properties of the measure.
Author Response
We thank reviewer 1 gratefully for your generous comments to improve the quality of our paper. Accordingly, we have made the necessary revisions in the manuscript and provided our responses below.
With respect to misclassification, we compared results from the 2018 and 2017 CHSA boundaries and found that that quintiles were fairly similar, suggesting that misclassification, if it exists, is minimal. Further, based on the fact that BCIMD quintile allocation is consistent with some healthy status metrics as shown in our Community Health Profiles, we conclude that misclassification is not an issue in our study. These comments have been added to the Data Limitations section.
We have added some sentences to educate readers on ecological fallacy in the Data Limitations section. However, the use of area-based deprivation index has been an acceptable and often relied upon practice in health equity surveillance in Canada and other jurisdictions, so we do not consider this to be a major issue.
We have added in a statement in the Results and Discussion sections that BCIMD scores can change over time, and that the scores are only valid within each Census cycle.
Reviewer 2 Report
Dear Author,
thank you for the opportunity of reading your manuscript. Follows some comments for improving the overall quality of your work.
Into the introduction, please give more emphasis to healthcare system current issues as well as to the main reasons at the core of measure and track health inequities. Some of the details presented into the Introduction about the available index (and measures) should be moved into the following sections (e.g., into the methodological ones). Reference to the extant literature are too poor. Moreover, some insights about the implemented methods and the achieved results should be presented into this section. Finally, Data Collection (in italic into the manuscript) is definitely not related to the introductory notes, but rather this sub-section must be moved into the methodological section.
The work also lacks a theoretical section in which the authors related the issue/topic under investigation to the extant literature and, therefore, with existing theories/frameworks.
Findings are not clearly presented and no results’ discussion has been provided. Therefore, it is not clear if and how the achieved results can be related to the number of previous studies on the topic. In a similar vein, no final remarks and/or theoretical, managerial, and policy implications (essential for works on this topic) have been provided.
Author Response
We thank reviewer 2 gratefully for your generous comments to improve the quality of our paper. Accordingly, we have made the necessary revisions in the manuscript and provided our responses below.
Into the introduction, please give more emphasis to healthcare system current issues as well as to the main reasons at the core of measure and track health inequities.
In our provincial jurisdiction, like elsewhere in Canada, we have a well functioning universal coverage of healthcare that provides health care services to all residents of British Columbia. However, there are certain subpopulations across demographic, geographic, and socio-economic groups that are disadvantaged and do not experience the same level of health access and health status as the rest of the population. Therefore, it is imperative for us to be able to examine and attempt to quantify the existing disparities for mitigation. This can be done by examining measures (indicators) stratified by various dimensions including the deprivation index.
Some of the details presented into the Introduction about the available index (and measures) should be moved into the following sections (e.g., into the methodological ones).
Thank you for this valuable suggestion. We have re-arranged the text contents according to your advice.
Reference to the extant literature are too poor.
Our study is a follow-up study of the development of various deprivation indices in Canada. It is not a primary study, but an applied adaptive research for developing a data product of deprivation index based on the theoretical development of Canadian Index of Multiple Deprivation by Statistics Canada. Therefore we believe that an exhaustive literature review is not intended or necessary for our study.
Moreover, some insights about the implemented methods and the achieved results should be presented into this section.
The implemented method from the previous referenced study is adequately described in our paper as an adaptation for generating a new data product for a different geographical configuration (ie CHSA), and appropriate references have been cited.
Finally, Data Collection (in italic into the manuscript) is definitely not related to the introductory notes, but rather this sub-section must be moved into the methodological section.
Thank you for this advice. We have moved the relevant data collection and analysis parts to the methods section.
The work also lacks a theoretical section in which the authors related the issue/topic under investigation to the extant literature and, therefore, with existing theories/frameworks.
We have added a theoretical underpinning of the development of the deprivation index as the summary measure of the various socio-economic variables that are correlated with the health inequities.
Findings are not clearly presented and no results’ discussion has been provided. Therefore, it is not clear if and how the achieved results can be related to the number of previous studies on the topic. In a similar vein, no final remarks and/or theoretical, managerial, and policy implications (essential for works on this topic) have been provided.
Results and Discussion sections have been re-organized and presented as advised.
Reviewer 3 Report
Comments
Line 36: The authors should define and describe horizontal equity and vertical equity.
Lines 38-41: The authors should mention the full range of administrative data disadvantages and advantages.
Lines 38-41: The authors should define access to healthcare and utilization of healthcare.
Line 44: How are inequities reflected in summary measures?
Line 44: The authors should define health equity and health equality.
Lines 48-54: The authors should describe the indices.
Lines 48-54: The authors should describe the advantages and disadvantages of the indices.
Lines 79-80: The authors should describe the characteristics and challenges.
Lines 83-96: The authors should discuss the availability of health and healthcare data.
Line 144: What does the blue arrow mean? Please clarify.
Lines 176-177: The authors should describe the health system characteristics.
Lines 259-272: Did the authors treat the “degree of urbanization” variable, as nominal (in the ANOVA)? The degree of urbanization is not a nominal variable.
Author Response
We thank reviewer 3 gratefully for your generous comments to improve the quality of our paper. Accordingly, we have made the necessary revisions in the manuscript and provided our responses below.
Line 36: The authors should define and describe horizontal equity and vertical equity.
We have defined disparity, equity and equality to set the context of our study. Horizontal and vertical equity lie outside the scope of this paper. Appropriate references are provided.
Lines 38-41: The authors should mention the full range of administrative data disadvantages and advantages.
We have added more information in this section, including some advantages and disadvantages of administrative data. In our study, administrative data are aggregated at the area-based geographic level for which the Census data compliments the socio-economic variables.
Lines 38-41: The authors should define access to healthcare and utilization of healthcare.
The scope of our paper is limited to population-level health status of the people associated with the socio-economic deprivation composite measures, and therefore, defining health care utilization would be out of scope.
Line 44: How are inequities reflected in summary measures?
We have added a theoretical underpinning of the development of the deprivation index as the summary measure of the various socio-economic variables that are correlated with the health inequities.
Line 44: The authors should define health equity and health equality.
We have defined disparity, equity and equality to set the context of our study. Appropriate references are provided.
Lines 48-54: The authors should describe the indices.
We have added more information about the indices in this section.
Lines 48-54: The authors should describe the advantages and disadvantages of the indices.
We may not need to explicitly mention advantages and disadvantages of indices as our paper considers indices useful for differentiating disparities in health. Appropriate references are provided. We have also provided some limitations to indices in the data limitations section.
Lines 79-80: The authors should describe the characteristics and challenges.
Thank you for this comment. We have added more information clarifying the characteristics and challenges, as per your suggestion.
Lines 83-96: The authors should discuss the availability of health and healthcare data.
In our provincial jurisdiction, like elsewhere in Canada, we have a universal coverage of healthcare that allows the collection of health and health care data from all residents of British Columbia. These data are available for the purpose of health research and surveillance.
Line 144: What does the blue arrow mean? Please clarify.
We have removed the blue arrow and instead put in the text that the dimensions are ordered according to the percentage of principal component variance explained. The percentage of variance decreases from the highest dimension listed on the left to lowest on the right.
Lines 176-177: The authors should describe the health system characteristics.
In our provincial jurisdiction, like elsewhere in Canada, we have universal coverage of healthcare that allows collection of health and health care data from all residents of British Columbia. These data are available for the purpose of health research and surveillance. We have added a description of our health care system in the Introduction.
Lines 259-272: Did the authors treat the “degree of urbanization” variable, as nominal (in the ANOVA)? The degree of urbanization is not a nominal variable.
This is a categorical variable. We have provided further explanation of this analysis in the Data Limitations section.
Round 2
Reviewer 3 Report
The authors improved the manuscript.